# Effects of Automotive Test Parameters on Dry Friction Fiber-Reinforced Clutch Facing Surface Microgeometry and Wear—Part 2

**DOI:** 10.3390/polym14091757

**Published:** 2022-04-26

**Authors:** Roland Biczó, Gábor Kalácska

**Affiliations:** Institute of Technology, Szent István Campus, MATE, Páter Károly u. 1., H-2100 Gödöllő, Hungary; kalacska.gabor@uni-mate.hu

**Keywords:** dry friction, hybrid composite, pin-on-disc, coefficient of friction, surface roughness, wear, activation energy, clutch

## Abstract

Coefficient of friction values, wear and surface roughness differences are revealed using pin-on-disc test apparatus examinations under three *pv* loads, where samples are cut from a reference, unused, and several differently aged and dimensioned, used, dry friction fiber-reinforced hybrid composite clutch facings. Tests are characterized by surface activation energy and separated into Trend 1, ‘clutch killer’, and 2, ‘moderate’, groups from our previous study. The results reveal that acceptable, 0.41–0.58, coefficient of friction values among Trend 1 specimens cannot be reached during high *pv* tests, though the −0.19–−0.11 difference of minimum and maximum *pv* results disappears when activation energy reaches 179 MJ. The maximum *pv* friction coefficient can decrease by up to 30% at working diameter due to clutch killer test circumstances, as 179 MJ surface activation energy is applied, while by moderate tests such losses can only be detected close to 2000 MJ energy values among small-sized facings. Besides that, Trend 2 specific wear values are the third of trend 1 results at inner diameter specimens. Compared to reference facing values, specific wear results at working diameter under maximum *pv* decrease by 47–100%, while increasing specific wear during lifetime can only be detected at the inner diameter of facings enduring clutch killer tests or that are small-sized facings. Among Trend 1 radial and tangential Ra delta results, inner diameter samples provide more decreasing surface roughness data, while by Trend 2 values, the opposite relation is detected. Apart from the effects of activation energy, mileage and driver profile, facing size and friction diameter influence is also revealed.

## 1. Introduction

Clutches are responsible for torque transmission in conventional automotive applications linking a rotating crankshaft—coupled to a power source—to a transmission system and to the wheels of the vehicle [1]. The most common type, the single disc clutch system, can be divided into sub-systems such as the flywheel, the clutch disc with hybrid composite friction facings and the clutch itself. Fiber-reinforced hybrid composites—most used materials as a friction surface in such applications—are the results of a century-long material development process [2].

However, being one of the mating surfaces in the frictional system, both properties and the torque transfer capacity of the facings change during their lifetime. Suppliers often designate mileage as the ‘best before’ characteristic. Though, dozens of other factors influence the performance of dry clutch facings. 

Furthermore, such complex hybrid composites consist of dozens of different components, making even their identification and characterization a complex task. These components are usually sorted into groups distinguished by their mechanical role and function, but many studies proved that they affect many different characteristics besides the targeted aspect. For instance, even reinforcing fibers can effectively influence tribological—friction and wear—properties [3] the same way binders are effective friction-modifiers [4]. Moreover, the fiber orientation of glass fiber composites has been found to have an influence on the friction and dry wear behavior [5]. Characterization is often challenged further by components being industrial secrets demanding a throughout research.

Therefore, in our recent studies [2], we introduced a novel material identification method to characterize a certain dry clutch hybrid composite friction material, with some of its components being an industrial secret. The study provided not only mechanical but also thermal characterization that allowed for the preparation of a complex material model, for example, in a finite element environment.

Furthermore, we continued to examine this material through its tribological properties—surface roughness and wear—during its lifetime after certain automotive tests—based on industrial best practice—described by surface activation energy values [6]. Surface activation energy was chosen to create a test intensity scale since during a dry friction automotive clutch engagement, assuming constant friction, the *pv* (pressure multiplied by velocity) value is proportionate to the dissipated thermal energy. This energy in the form of surface activation energy transforms the surface microgeometry and therefore leads to wear phenomena, temperature rise and entropy changes associated with interface material transformation [7]. Friction and radial disc motions contribute to noises and vibrations as well [8].

However, to determine the reliability—the capability of further utilization in the clutch application—of the preliminary aged discs regarding their tribological behavior and performance, additional examinations are required. The so-called pin-on-disc test apparatus provides an effective way to carry out such research. Therefore, since ceramic materials are also utilized as friction facings in high load applications, Poser et al. [9] utilized this apparatus to compare Al_2_O_3_-based ceramic composite clutch facing tribological performance with monolithic alumina results. They concluded that the improved friction coefficient stability was the result of TiN particles imbedding into the friction surface. In the meantime, since woven hybrid composites are still the most widely used automotive dry clutch friction materials, Bezzazi et al. [10] used pin-on-disc test setup to acquire friction coefficient and wear values of organic clutch facings. They evaluated fading resistance and stability of the coefficient of friction. Contrastingly, from pin-on-disc friction coefficient results, Senatore et al. [11] created an artificial neural network capable of predicting the influence of main sliding parameters. Both brake and automotive clutch materials were examined by them and sliding acceleration was found to be an important parameter. Pin-on-disc frictional models were found to be helpful during the observation of a transition from mild to severe wear under specific load conditions governed by hardness, normal load, surface roughness, sliding velocity and temperature through time as Fernandes et al. concluded [12]. On the other hand, these test devices only allow one to investigate the starting and final conditions of friction behavior with the amount of friction debris. The mechanism of the transition though can only be estimated. The influence of sliding conditions on the stability of multi-layer friction films was investigated by Fernandes et al. [13]. They concluded that removing wear debris results in increasing friction level and reducing wear rates. Nirmal [14] used pin-on-disc test results of bearing applications to determine the effects of fiber-reinforced polyester composite aging on friction coefficient reduction. It was revealed that aging solutions with lower kinematic viscosities revealed lower friction coefficients, whereas the friction and wear response of 3D-printed polymer composites, namely, acrylonitrile butadiene styrene (ABS), in the presence of graphite powder lubricant was examined on pin-on-disc apparatus by Keshavamurthy et al. [15]. They demonstrated decreased coefficient of friction as a result of the graphite addition in the ABS matrix. In the meantime, Mrówka et al. [16] performed frictional influence characterization of metallurgical production waste products as fillers in silicone composites prepared by casting. Pin-on-disc tests revealed that zinc waste can reduce abrasive wear and therefore environment pollution by reusing this production waste as fillers.

On the other hand, a pin-on-disc setup has its limitations, and considering production tolerances, for instance, describing the tribological behavior of the whole friction facing based on a small area pin result, is of course only an approximation. Therefore, Hoic et al. [17] designed a special pin-on-disc (disc-on-disc)-type tribometer to be able to characterize static and dynamic friction behaviors of whole dry friction clutch facings among other friction pairs. The main parameters of their experiments using the machine are the normal force, slip speed, and temperature. Furthermore, this device, the disc-on-disc apparatus, was utilized by Hoic et al. [18], revealing that each change of friction interface temperature level induces transient behavior. 

Another aspect that a basic pin-on-disc setup lacks is the high-temperature environment a clutch facing experiences in a vehicle. Therefore, Humphrey et al. [19] recreated thermal tribodynamic conditions via a special pin-on-disc tribometer supplied with a heating system: a special copper disc. Rising contact temperature and increasing slip speed were found to be responsible for decreasing the kinetic coefficient of friction of the clutch lining.

Despite many investigations, concentrate on the behavior of different composites among pin-on-disc test circumstances just out of the production line or phase, there are not many studies covering tribological behavior at certain stages of the whole lifetime of the composite. Automotive dry clutch investigations in particular lack examination of the effects of real life or test track usage in actual vehicles, as well as comparison of tribological performance of the same material with different preliminary activation energy levels of the friction surface.

Therefore, our aim is to continue revealing the tribological, lifetime-affected aspects of dry friction hybrid composite clutch facings that we began in our recent study this time under prescribed energy loads on pin-on-disc test apparatus, concentrating on friction, wear and surface roughness values, possible correlations between them and effects on each other. Section 2 highlights the results of our recent studies regarding the facing properties and field-test tribological behavior. Section 3 introduces coefficient of friction, wear and surface roughness results after laboratory pin-on-disc tests on specimens cut from the previously tested facings revealing tribological performance during lifetime.

The results are still aimed at dry clutch finite element contact model development and refinement. The goal is to determine the applied energy dependence of tribological properties and factors. 

## 2. Materials and Methods

Complex hybrid composite materials need throughout investigation in order to sum up our thermomechanical characterization and modeling method with tribological aspects detailed in our previous studies [2,6]. Figure 1 provides illustration.

Mechanical and thermal property evaluation tests are for thermomechanical characterization. Surface roughness and wear measured after automotive tests and then pin-on-disc investigations are responsible for tribological behavior characterization.

### 2.1. Tested Materials

The dry clutch friction material investigated in this study was a conventional dry friction hybrid composite facing material identified and characterized mechanically and thermally in one of our recent studies [2]. It was manufactured by the scatter wound process. Figure 2 illustrates the main components of this material and its location in the transmission system.

Mechanical stiffness matrix parameters and thermal properties of this woven fiber-yarn- (glass fiber with aromatic polyamide, copper, and poly-acrylic-nitrile (PAN)) reinforced friction material were determined. Table 1 details the results of the study that grants an effective reference and a novel guidance for material identification methods for similar complex materials

### 2.2. Real-Scale Field Testing before Pin-On-Disc

The dry friction clutch facing material characterized as a first step was then examined from a tribological point of view in a novel way as Part 1 of this article details [5]. Surface roughness and wear values after automotive tests run with vehicles equipped with clutch facings made from this previously identified friction material were evaluated. In order to prepare with proper settings and samples for the pin-on-disc tests detailed in this article, a comprehensive review of the field-test categories is required. Therefore, Table 2 provides an overview of the test matrix covering not only mileage but many other parameters. Abbreviations used for test names are: T: test bench, H: highway, V: vehicle, C: city, VT: vehicle + trailer, R: testing ring (test track), RS: testing ring + hill start). Surface activation energy calculations are detailed as well in Part 1 of this article and are used to evaluate wear and surface microgeometry results [6]. Among the facings, only dimensions (Ø228/Ø160/3.5 mm; Ø240/Ø160/3.8 mm; Ø240/Ø155/3.8 mm outer/inner diameters and thickness, respectively) differ besides preliminary or initial wear states. The results show that wear and surface microgeometry aspects of fiber-reinforced hybrid composite dry friction clutch facings on two different facing diameters and in two directions vary during lifetime according to surface activation energy rather than mileage. Along the increasing activation energy scale, wear values increased according to two different trends, sorting tests into two main groups, namely, ‘clutch killer’ and ‘moderate’, promoting drivers’ handling of the clutch as a main factor. Facing geometry properties such as thickness, outer and inner diameter were recognized as potential factors governing dry clutch tribological performance during its lifetime as well. Surface roughness value trends varied similarly, highlighting the effects of the same factors [6]. 

However, for throughout tribological characterization besides surface roughness and wear, aspects of coefficient of friction requires investigation. The pin-on-disc test method provides an opportunity to carry out such experiments.

### 2.3. Pin-On-Disc Test Method

Tribological examination was carried out via pin-on-disc tests to evaluate effects of certain *pv* values on surface microgeometry, wear and coefficient of friction of dry friction hybrid composite clutch facing. So-called pins were cut from those facings that endured those preliminary test conditions that are detailed in the previous chapter along with facings fresh out of production to provide reference values. As we stated in one of our recent studies [2], abrasive water jet machining turned out to be an effective tool for sample creation for our investigations. As Figure 3 illustrates, this method allowed us to use precisely cut Ø7 mm diameter samples. These were then glued with universal superglue to metal ‘holder-pins’ for fixating them on the tribo-apparatus.

On the pin-on-disc test apparatus, continuous friction sliding rises between the surfaces of the fixed Ø7 mm hybrid composite test samples and the rotating disc made of GG25 cast iron. The examined pin-on-disc tribological system works according to standard pin abrasion testing (ASTM G-132): without lubrication, among dry abrasive circumstances. Ambient temperature is the room temperature.

Figure 4 shows the main units of the test system: (1) table; (2) disc; (3) positioning system; (4) loading system; (5) electric motor with speed setting rotating the disc (only belt drive visible).

During the tribology, the test parameters are the sliding speed, the timeframe of the procedure, the normal load, the friction radius, the surface roughness of the disc, the environmental temperature and relative humidity. Awaited results are the dynamic friction coefficient and wear as vertical displacement of the specimen holder with deformation of the specimen neglected. 

The surface roughness of the GG25 disc is evaluated with Mitutoyo SJ-201P application. Its value should lie in the range of Ra 4–7 µm during all test runs as it is the production value of a flywheel friction area. Rotational velocity minimum value is 120 1/min, maximum value is 240 1/min. Friction radii were 20 mm and 40 mm. Therefore, the sliding speed values calculated are 0.5 or 1 m/s. Normal load was applied by dead weight attached to the apparatus creating 31 and 68 N force loads (F_n_). From the given parameters, three *pv* levels were reached, namely, minimum, ~0.4 MPa∙m/s, medium, ~0.9 MPa∙m/s and maximum, ~1.8 MPa∙m/s. 

As Figure 5 illustrates, F_s_ friction force rises parallel with the friction surface while F_n_ = F_z_ normal force acts. F_x_ and F_y_ are force components parallel to the friction surface. 

Therefore, the dynamic friction coefficients are determined according to Equation (1):(1)$$\mu =\frac{F_{s}}{F_{n}}=\frac{\sqrt{F_{x}^{2}+F_{x}^{2}}}{F_{z}}$$

Being a torque transmission system, during actuation the clutch is required to reach a certain synchronized rotational velocity to fully transmit the load from the engine towards the transmission. During this torque build-up phase, as illustrated via the graph in Figure 6, the calculated *pv* number describing the load of the clutch facing on its working diameter is the result of a decreasing velocity difference (dv) between the contact pairs (engine shaft velocity of the flywheel versus increasing velocity of the clutch facing) multiplied by the increasing pressure load (p) on the facing as it travels along its cushion deflection displacement. The illustrated simplified situation assumes that constant acceleration of the transmission shaft the clutch disc is connected to an automotive transmission system that has a Ø228/Ø160 mm (outer/inner) diameter friction facing and 1800 RPM engine speed at the beginning of the actuation. The *pv* value dependence on diameter is also highlighted, along with the minimum, medium and maximum *pv* values selected for our measurements on the pin-on-disc (PoD) apparatus. 

As Table 2 details, samples differ regarding the facing geometry (diameters and thickness), the specific diameter they are cut from and the surface activation energy due to different test types (intensity of actuation, driver profile etc.) and the mileage the vehicles they were built in ran. 

Along the facing diameter, not only pressure but also temperature distribution changes, as Meng and Xi reported after calculations and using temperature-sensing technology [20]. This explains the two different diameters—working and inner—the samples are cut from. During the lifetime of the clutch, due to residual deformations and wear, the ideal, initial parallel position of the mating surfaces changes as well, and higher wear values are awaited at the inner diameter after a considerable amount of activation energy has been applied.

Furthermore, despite the fact that three *pv* values were determined to describe the phases of the clutch actuation procedure, not all were applied to every type of sample Table 2 describes. Moreover, not all inner diameter or working diameter samples were tested. The selection of them was determined based on their wear and roughness results after automotive tests detailed in Part 1 of this article considering the two utilization profiles, ‘clutch killer’ and normal usage as well.

## 3. Dry Friction Hybrid Composite Tribological Behavior Investigation Results on Pin-On-Disc Apparatus

The aim of the tribological investigation was to compare tribological performance, wear and surface microgeometry characteristics of differently aged dry friction clutch facing surfaces among laboratory circumstances created via the pin-on-disc (PoD) apparatus. Different age stands for different surface activation energy values that the facings had endured in automotive tests. 

Samples of one of the PoD test sample categories detailed in Table 2 were cut from a facing right after production, to serve as a reference for the tribological performance. All the other samples were cut from facings that endured different activation energy load during their lifetime. All samples are common regarding their material, in which the long fiber reinforcement consists of glass fiber, aromatic polyamide, copper, and poly-acrylic-nitrile (PAN), while the matrix is an epoxy-based resin: a short fiber-reinforced composite itself, including a melamine-modified epoxy phenol resin filled with compounded rubber and further components: sulfur, aromatic polyamide as short fiber reinforcement, and other filler materials.

The goal was to determine factors that influence facing material tribological performance and behavior sensitivity during a clutch’s lifetime. Comparing relative result values helps to highlight dependence on the different parameters. Therefore, results such as wear and surface roughness difference (compared to results after preliminary field tests) are normalized to activation energy.

### 3.1. Coefficient of Frcition Values

During the pin-on-disc tests, online signals of strain gauges that measure the magnitude of the forces applied to the test specimen were detected continuously. Coefficient of friction (CoF) values were calculated from these forces as Section 2.3 details. Figure 7 provides an example of coefficient of friction characteristic from the online measurement. 

Two parameters were determined for each specimen’s *pv*-level measurement: the maximum CoF and the steady state CoF. From each measurement, the former is the highest coefficient of friction during the running-in phase, the latter is the mode of all the coefficient of friction values calculated. Steady state friction phase is considered from the point when the difference between calculated steady state and measured coefficient of friction values is lower than 0.001. 

One of the most important criteria regarding automotive clutch facings is their tribological behavior during the time period the driver slips the clutch. The longest period it can occur is when the car is in heavy, slow-moving traffic or starts moving upwards on a hill. This condition is considered for the coefficient of friction values as the results of the first 10 s of the measurement are considered for the evaluation.

Coefficient of friction criteria are determined based on actuation number during bench tests—based on best practice in the automotive industry. Converting them into surface energy values—as our previous study [6] detailed—provides coefficient of friction criteria values with a minimum of 0.41 and a maximum of 0.58. Facings operating with coefficients outside of this range are not considered applicable for proper torque transmission anymore. However, facings after production should operate within the 0.31–0.51 range regarding their coefficient of friction. 

Figure 8 details the results of the steady state coefficient of friction value evaluation from pin-on-disc (PoD) test data by the so-called ‘clutch killer’ trend recognized in our recent study [6], while Figure 9 illustrates the results of the PoD by ‘moderate’ trend field-test samples. Both are illustrated along non-linear Joule scale of preliminary field-test activation energy values. The results of different *pv* values (min, med, max) are differentiated by color, while the results of different diameters (di: inner diameter, dw: working diameter) the specimens are cut from are distinguished by color shades. The results are based on three repetitions. 

From the results comparing Figure 8 and Figure 9, the following can be stated:Clutch killer sample coefficient values fall into the 0.41–0.58 criterium range mainly by inner diameter samples only under minimum and medium *pv* loads, while moderate values perform in the acceptable range under medium and maximum *pv*.Regardless, the cutting diameter ‘clutch killer’ sample results under high *pv* fall out of the acceptable, 0.41–0.58 coefficient of friction range.Increasing *pv* from 0.4 to 1.8 MPa∙m/s clutch killer CoF values (belonging to the same di or dw diameter) causes decreases of 0.11–0.19 by low surface energy tests, while this difference is not present anymore, as field-test energy values are as high as 179 MJ.Increasing *pv* from 0.4 to 1.8 MPa∙m/s moderate CoF values (belonging to the same di or dw diameter) causes an increase of ~0.05, except for the small-sized (S) facing specimens.VTRS results in particular highlight that the coefficient of friction decreases from inner diameter to working diameter, reflecting the fact that the more aged a facing, the higher the prescribed criteria values are—see change in Figure 8 and Figure 9: minimum value from 0.31 to 0.41; maximum value from 0.51 to 0.58; and working diameter at a lower surface energy probably has more wear and a lower coefficient of friction.At higher activation energy values, ‘clutch killer’ inner diameter sample coefficients are lower than the working diameter sample coefficient by ~0.07, reflecting the phenomena that towards the end of the facing’s lifetime, due to deformations in the clutch, the inner diameter starts to endure higher thermal loads, which leads to poorer tribological performance.

The results, on the other hand, clearly show that the dependence of the coefficient of friction on surface activation energy during lifetime is influenced by several factors. Compared to mileage, user profile, geometry and friction diameter play a more critical role.

### 3.2. Wear Values

Wear values were also monitored during the pin-on-disc test and evaluated from the signal of a calibrated height sensor measuring the vertical displacement of the pin holder. Though, this way, deformation values were also detected, due to the small size of the test sample, they were negligible. Figure 10 provides an example of the results of wear detection (linear decrease in thickness (mm)) on different *pv* values for the same specimen category along the friction path (s). 

Due to the different activation energy levels that originated from field experiments [6], test parameters and measurement length wear values are normalized to the activation energy applied during the pin-on-disc test. This way, the specific wear values to 1 kJ activation energy at certain stages of the lifetime of the material are compared. The results are illustrated in Figure 11 and Figure 12 for the two different trends recognized in our previous study [6]. The results of different *pv* values (min, med, max) are differentiated by color, while the results of different diameters (di: inner diameter, dw: working diameter) the specimens are cut from are distinguished by color shades. The results are based on three repetitions.

The following conclusions can be drawn examining the results of the specific wear measurements of PoD following previous field tests (Table 2):The initially higher wear values at inner diameter [6] makes the material more sensitive to high *pv* values, as VRS results suggest: 0.03 mm/kJ reached and even exceeded under medium and maximum *pv*.Mileage also has an influence on the *pv*-sensitivity, as TH results higher than 0.02 mm/kJ suggest, though it is not as strong as the assumed intensive aging effect typical at inner diameter, as VRS results highlighted.Comparing results with the same *pv* value within the same test specimen category, more intensive wear at inner diameter can be detected not only by clutch killer (0.005–0.007 mm/kJ difference at low mileage and energy, 0.015 mm/kJ by more than 100,000 km mileage field test or at the end of surface activation energy scale) but also by moderate results (0.02 mm/kJ by samples from PoD following less than 60,000 km test, 0.01 mm/kJ by samples from PoD following 150,000+ km test).Based on the VTC results, the mentioned relations can also be the consequence of the fact that the facings were field-tested in a vehicle with a trailer, meaning increased vehicle weight and clutch engagements in non-optimal contact positions regarding the clutch friction surface.VTRS PoD test results can also be the consequence of the smaller diameter of the original facings. Higher wear rate at lower mileage was also detected merely from the field test compared to the other facings [6].City driving conditions (see C in abbreviations) with their extra energy load due to frequent shifting and shorter cool down opportunities created similar results to the ones enduring pin-on-disc after test bench (T) conditions

Examining Figure 12, moderate test facings pin-on-disc wear results highlight the effects of tests run on test tracks, in vehicles driven by professional (VR), and high mileage city tests, where the customer’s driving style determined the long-term wear values (VC), and the following can be seen:Compared to clutch killer results (0.01–0.03 mm/kJ), moderate specific wear values at inner diameter (0.01 mm/kJ) are significantly lower except for small-sized (S) facing.However, the effect of driver profile does not seem to be so significant among these specific values.By more or less the same mileage (50,000–60,000 km) smaller diameter, VR-tested facing (size S) showed higher specific wear values than greater diameter facings by 0.01–0.015 mm/kJ.VC-specific wear results suggest that if the system was applied with loads from non-professional drivers, but for a long time (mileage), specific wear values would be similarly moderate (below 0.015 mm/kJ) as those caused by professional drivers.

Overall, conclusions from specific wear results are that:Though the more energy applied, the higher the specific wear values, the exact amount is highly influenced by driving styles and the rapidity of energy load build-up at vehicle starts, highlighting the significance and effects of aging.Interestingly, higher *pv* values created lower specific wear results.Difference of specific wear values at different diameters is more significantly present by lower *pv* values or by higher mileage.The more intensive the preliminary test, the more homogeneous the specific wear spectrum, regardless of the *pv* values as well. This can be the result of the fact that, during its lifetime, the friction surface transforms after each and every engagement and a different surface is created after repeated contacts with the counter surface, modifying the latter as well.

Regardless of specific values, parameters still have influence on wear of the examined dry friction hybrid composite clutch facing. 

### 3.3. Surface Roughness Values

Examining surface roughness values of friction facings after PoD tests on specimens from a reference unused facing and facings that endured different preliminary field tests gives us another opportunity to compare the tribological behavior marks during the lifetime—along the non-linear Joule scale—of the examined dry friction hybrid composite clutch facing material. Therefore, surface roughness measurements were carried out utilizing the MarSurf surface roughness measurement device equipped with a PHT 350 header. During these measurements, measuring length was Lt = 4.8 mm, and arithmetic average roughness, Ra, ten-spot average roughness, Rz, and maximum height, Rmax—all three according to JIS B0601-1994—were evaluated. Figure 13 illustrates the directions in which examinations were carried out.

Result values were then compared to surface roughness values before PoD and surface roughness difference values calculated for all specimens in Ra, Rz and Rmax. The same way wear values were treated and evaluated due to different activation energy levels [6], test parameters and measurement length, surface roughness difference values are normalized to activation energy applied during the pin-on-disc test. This way, the specific surface roughness change (specific dR) values to 1 kJ activation energy at certain stages of the lifetime of the material are compared. The results are illustrated in Figure 14, Figure 15, Figure 16 and Figure 17 for the two different trends recognized in our previous study [6]. The results of different *pv* values (min, med, max) are differentiated by color, while the results of different diameters (di: inner diameter, dw: working diameter) the specimens are cut from are distinguished by color shades. The results are based on three repetitions.

Examining radial direction Ra delta values, the following can be stated:Reference measurement shows PoD-induced decreased roughness with slightly more (0.4%/kJ) decrease on working diameter along the activation energy scale.This relation between inner and working diameter sample results changes for the opposite by almost all the samples with preliminary tests, except for S size facing samples.Most of the Ra values decreased, except for VTC inner diameter specimen results under minimum *pv* and VTC inner diameter specimen under medium *pv* at high mileage and energy providing positive, more than 5%/kJ dRa, values. These reflect, that the surface is torn up since resistance is low due to preliminary test consequencesInterestingly, under medium *pv*, increase occurred only at the VTRS test facing working diameter specimen—reflecting on the effects of tests run with a trailer.Among Trend 1 results, contrary to reference samples inner and working diameter surface roughness delta relation, inner diameter samples provide more decreasing surface roughness data, while by Trend 2 values, the reference relation remains, meaning that deformations causing increased inner diameter tribological performance loss can be prevented by proper handling of the clutch by the driver.Under maximum *pv*, Ra-increase occurred only by samples from S size facing and high preliminary activation energy facings among Trend 2 results.Otherwise, roughness delta absolute value decreases with increasing *pv* by ~0.5–1%/kJ.

Radial direction Trend 1 surface roughness delta changes the same way along the non-linear Joule scale regarding Rz and Rmax values. The same can be observed from Trend 2 results. Therefore, radial direction Trend 1 and Trend 2 Rz and Rmax results are illustrated in Appendix A, Figure A1, Figure A2, Figure A3 and Figure A4. The results are based on three repetitions.

In tangential direction, all three roughness delta values behave the same way along the Joule scale. Therefore, Figure 16 and Figure 17 illustrate only the Ra delta values along the non-linear Joule scale for Trend 1 and Trend 2 specimens, respectively, while Trend 1 and Trend 2 Rz and Rmax results are illustrated in Appendix A Figure A5, Figure A6, Figure A7 and Figure A8.

From the results that are based on three repetitions, the following can be concluded:In the tangential direction, the reference measurement shows decreasing specific roughness values of 2–4%/kJ after PoD following field tests. The effect of diameter is highlighted illustrating that inner diameter roughness decreases more significantly, with ~2%/kJ difference.This relation in tangential direction does not hold up along the Joule scale by Trend 1 results but remains more or less true regarding Trend 2 values.Trend 1 specimens cut from small-sized (S) facings that endured testing with a trailer (VT in abbreviations) provide a 1–7%/kJ increase in Ra at minimum and medium *pv*—see VTC. Higher, above 3700 MJ, surface energy samples in particular provide a 6–7%/kJ increment both by inner and working diameter samples.Among Trend 1 specimens, minimum and medium *pv* usually leads to increasing roughness, while maximum *pv* creates more smooth surfaces by decreasing Ra.Trend 2 roughness changes after PoD are more uniform regarding their direction. Mainly decreasing values are measured.

Similar to coefficient of friction and wear values, surface roughness examination highlights the effects of parameters governing the tribological behavior of dry friction fiber-reinforced hybrid composite clutch facings. This means that the applicability of such a facing cannot be judged simply by mileage as best practice suggests. 

## 4. Conclusions 

Following aging through different activation energy automotive best practice field tests of previously identified and thermomechanically characterized fiber-reinforced hybrid composite dry friction clutch facings, our recent study examined their tribological behavior regarding friction coefficient, specific wear and specific surface roughness changes among laboratory circumstances. Pin-on-disc tests were conducted under three different *pv*-levels, namely, minimum, ~0.4 MPa∙m/s, medium, ~0.9 Mpa∙m/s, and maximum, ~1.8 MPa∙m/s, according to the ASTM G-132 standard. Testing samples were cut via abrasive water jet machining from the inner and working diameter of those dry friction facings. Separation of preliminary tests into Trend 1, ‘clutch killer’, and Trend 2, ‘moderate’, groups was carried on from our previous article for the current study. Evaluation of results confirmed our hypothesis that activation energy and driver profile are the main parameters governing the tribological behavior of dry friction facings. The effect of further factors, such as facing geometry and friction diameter, were revealed as well.

Examining coefficient of friction results, the following can be stated:‘Clutch killer’ samples provided acceptable 0.41–0.58 values only with the inner diameter group samples under 0.4 and 0.9 MPa∙m/s *pv* loads. ‘Clutch killer’ sample results under high *pv* already fall out of the acceptable coefficient of friction range.‘Moderate’ specimens performed within the acceptable coefficient range by the working diameter group under 0.9 and 1.8 MPa∙m/s *pv*.The −0.19–−0.11 difference between ‘clutch killer’ coefficients belonging to 0.4 and 1.8 MPa∙m/s low-energy test samples is not present among test samples with field-test energy values as high as 179 MJWith increasing *pv* from 0.4 to 1.8 MPa∙m/s, moderate CoF values (belonging to the same di or dw diameter) increase by ~0.05 except for the small-sized (S) facing specimens.Maximum *pv* friction coefficient can decrease by up to 30% at working diameter due to clutch killer test circumstances as early as 180 MJ surface activation energy is applied, while by moderate tests such losses can only be detected close to 2000 MJ energy values among small-sized facings.

Specific wear results revealed the following:Compared to reference facing values, specific wear results at working diameter under maximum pv decrease by 47–100%, while specific wear during lifetime increases only at the inner diameter of facings enduring clutch killer tests or that are small-sized facings.Compared to clutch killer results (0.01–0.03 mm/kJ), moderate specific wear values at inner diameter (0.01 mm/kJ) are significantly lower, except for small-sized (S) facing, which showed higher specific wear values than greater diameter facings by 0.01–0.015 mm/kJ.Trailer as increased vehicle weight causes higher inner diameter wear: 0.005–0.007 mm/kJ difference by low-energy specimens, 0.015 mm/kJ difference by high-energy specimens compared to working diameter results.Mileage also has an influence on the *pv*-sensitivity, as TH results higher than 0.02 mm/kJ suggest.Inner diameter preliminary severe wear conditions [6] at VRS results leads to at least 0.03 mm/kJ specific wear under medium and maximum *pv*.Difference between inner and working diameter specific wear increases along the energy scale and/or by higher mileage tests.

From specific surface roughness change values, we concluded the following:Compared to reference facing values, all values decrease during lifetime Though greater than 100%, decrease is more likely to happen at working diameter.Among Trend 1 radial and tangential Ra delta results, inner diameter samples provide more decreasing surface roughness data, while by Trend 2 values, the opposite relation is detected. In tangential direction Trend 2, roughness changes after PoD are more uniform regarding their direction, mainly decreasing values are measured.Roughness delta absolute value decreases with increasing *pv* by ~0.5–1%/kJ.Positive, more than 5%/kJ radial direction, Ra delta showed torn up surfaces by samples from tests with a trailer, but under maximum *pv*, Ra-increase occurred only by samples from S size facing (1–7%/kJ increase in tangential direction) and high preliminary activation energy facings among Trend 2 results.Activation energy alters the relation of inner and working diameter radial direction Ra delta (0.4%/kJ more decrease on working diameter) for the opposite except for S size facing samplesInner diameter tangential roughness decreases more significantly, with a ~2%/kJ difference.

To conclude, the friction coefficient differences of ‘clutch killer’ and ‘moderate’ samples clearly reflect on the effect of the clutch-operating driver, since thermally induced deformations can occur by lower surface energy values regarding lifetime in case of misuses harming the working diameter tribological performance. During its lifetime, the friction surface transforms after each and every engagement and a different surface is created after repeated contacts with the counter surface, modifying the latter as well. As a consequence, the more intensive the preliminary test, the more homogeneous the specific wear spectrum, regardless of the *pv* values as well.

The effects of the facing size is highlighted by the opposite behavior of small-sized facing samples when load changes from medium to maximum *pv*. 

Different performance of inner and working diameter (due to thermal load induced deformations) is also highlighted by a 0.07 coefficient of friction difference or the higher specific wear and radial direction roughness losses.

Further investigations will focus on examining parameter interactions as well to fully characterize automotive dry clutch hybrid composite facing tribological behavior during lifetime. The final aim remains to extend the prediction capability of dry friction contact models currently utilized in the automotive industry coupled with the thermomechanical and tribomechanical analyses considered.

## Figures and Tables

**Figure 1 polymers-14-01757-f001:**
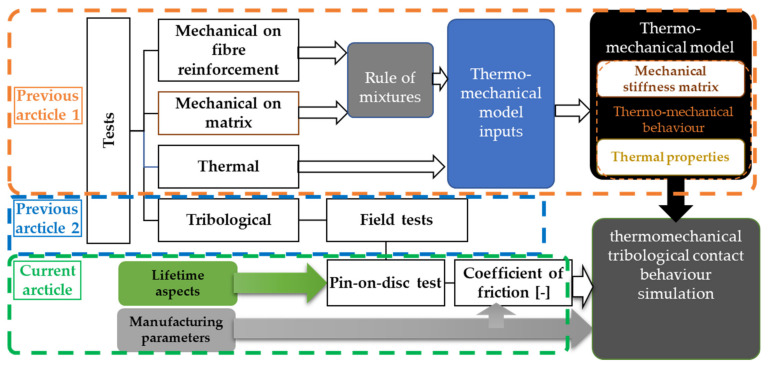
Thermomechanical characterization and modeling method with tribological aspects.

**Figure 2 polymers-14-01757-f002:**
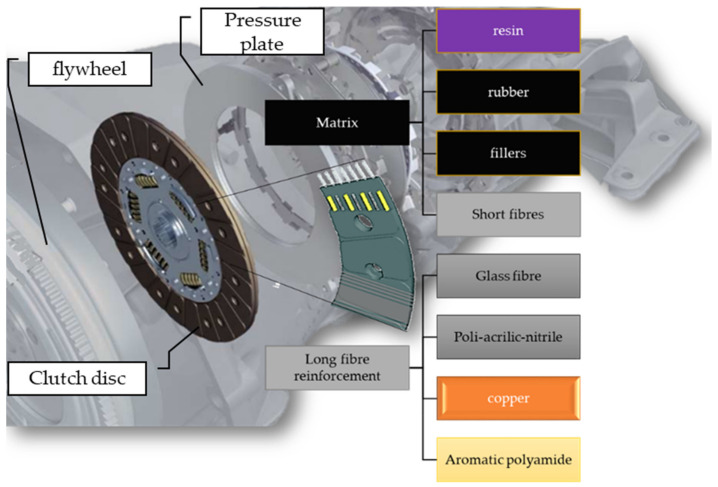
Components of the facing examined in our studies and its location in the transmission.

**Figure 3 polymers-14-01757-f003:**
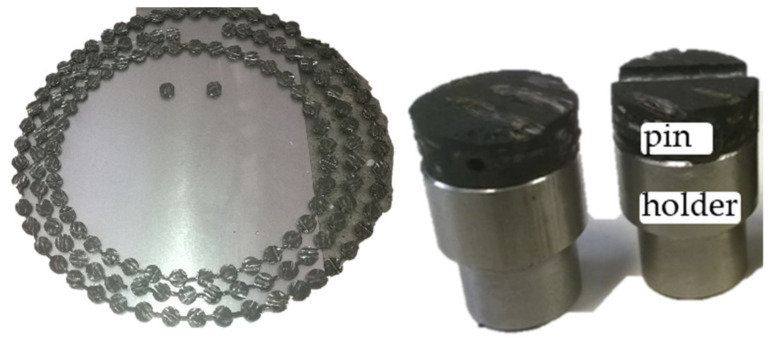
Samples cut from the facing and glued to metal holder-pins for fixation.

**Figure 4 polymers-14-01757-f004:**
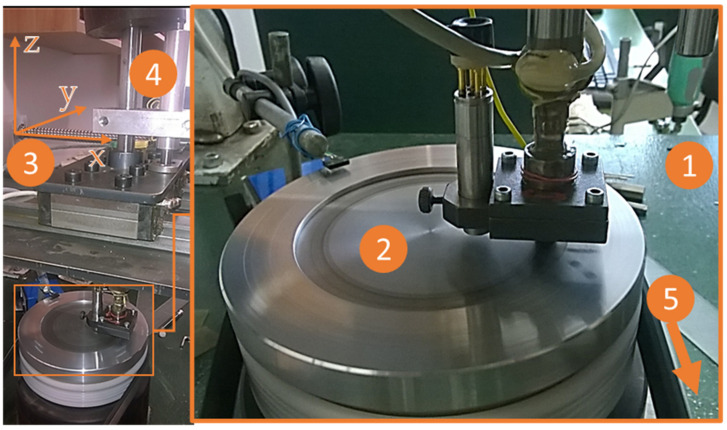
Main units of the pin-on-disc test system.

**Figure 5 polymers-14-01757-f005:**
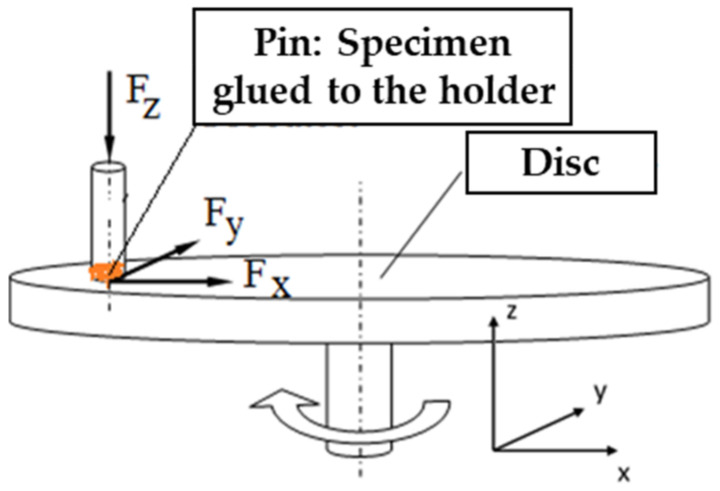
Schematic illustration of load components during the pin-on-disc test.

**Figure 6 polymers-14-01757-f006:**
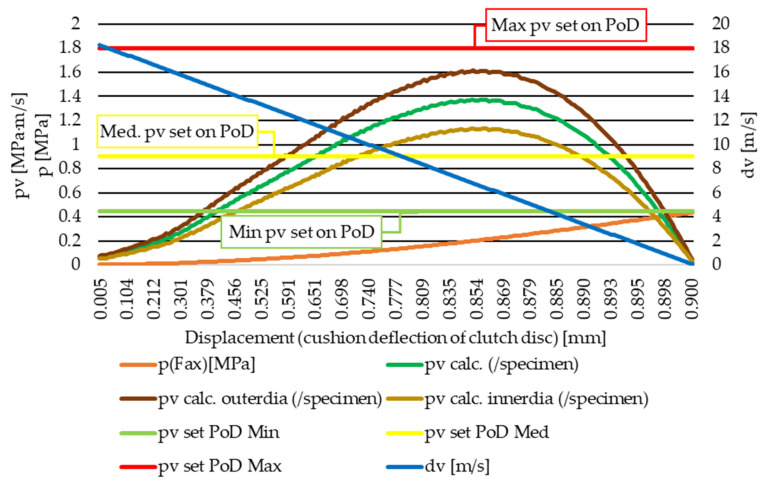
Example of calculated pv value change during clutch actuation along the facing deflection and the three set *pv*-values for pin-on-disc tests.

**Figure 7 polymers-14-01757-f007:**
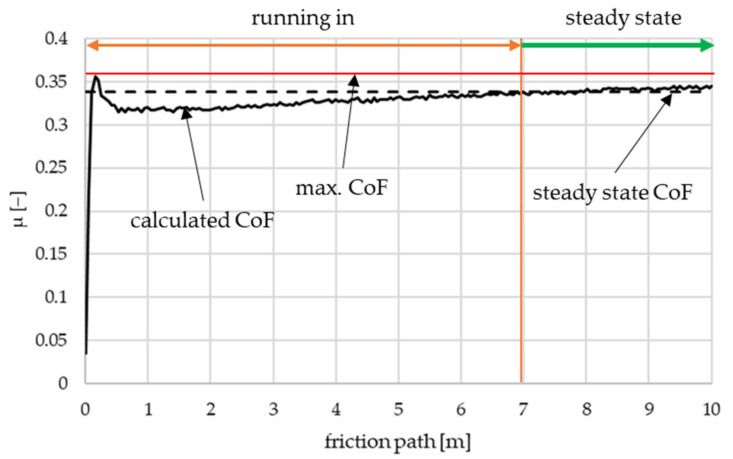
Example of coefficient of friction graph.

**Figure 8 polymers-14-01757-f008:**
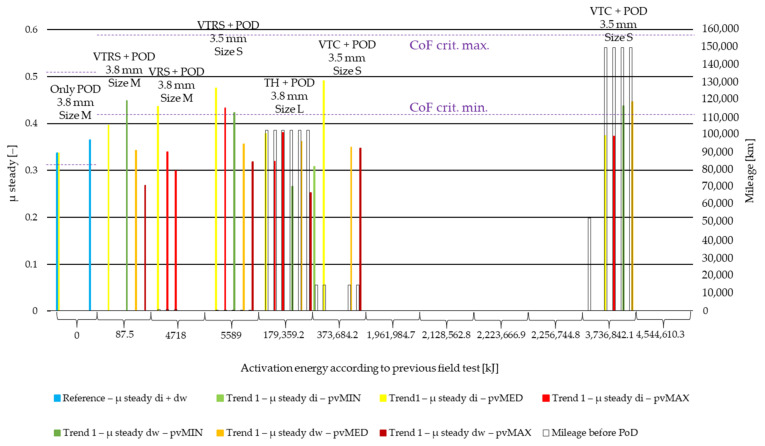
Steady state CoF values of PoD samples from ‘clutch killer’ field-test facings.

**Figure 9 polymers-14-01757-f009:**
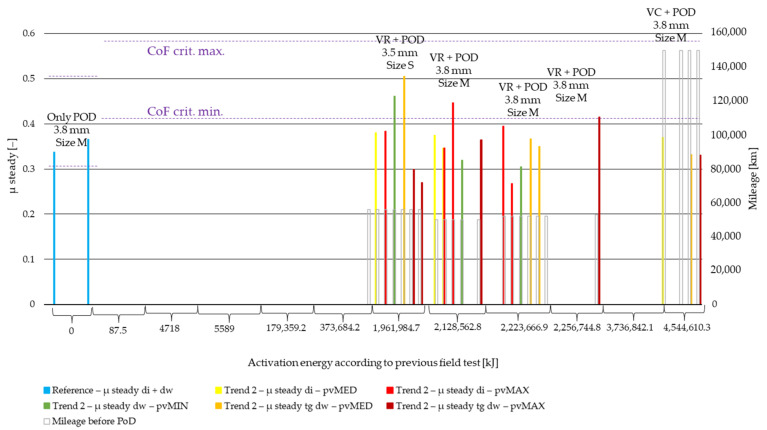
Steady state CoF values of PoD samples from ‘moderate’ field-test facings.

**Figure 10 polymers-14-01757-f010:**
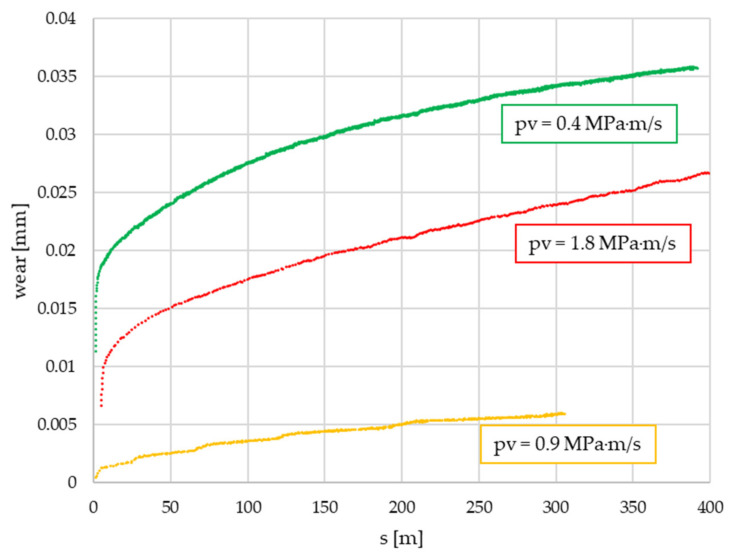
Wear values during the PoD test of a specimen category.

**Figure 11 polymers-14-01757-f011:**
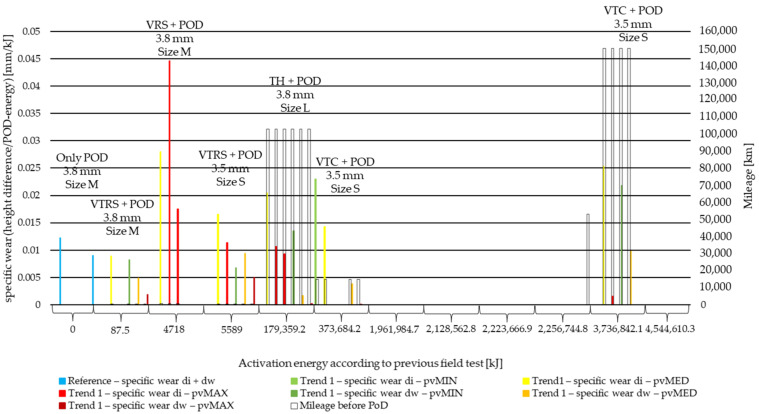
PoD specific wear values of cut material samples from inner and working diameter after minimum, medium and maximum *pv* level pin-on-disc tests with previous field-test (Table 2) mileage along Joule scale—Trend 1: clutch killer load cases: VTRS, VRS, TH, VTC.

**Figure 12 polymers-14-01757-f012:**
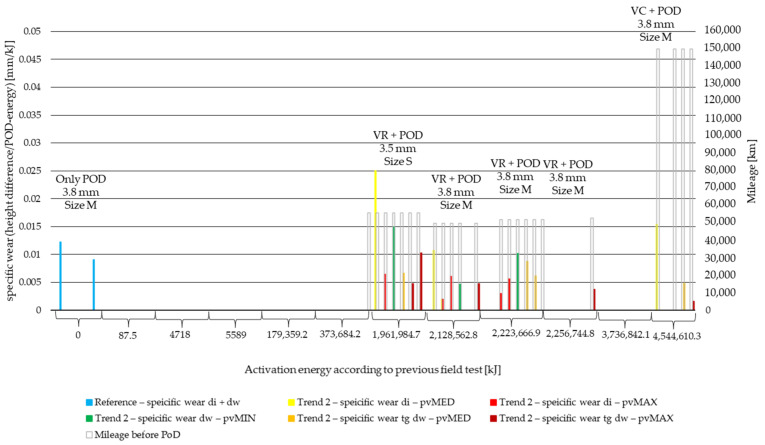
PoD specific wear values of cut material samples from inner and working diameter after minimum, medium and maximum *pv* level pin-on-disc tests with previous field-test (Table 2) mileage with mileage along Joule scale—Trend 2: moderate load cases: VR, VC.

**Figure 13 polymers-14-01757-f013:**
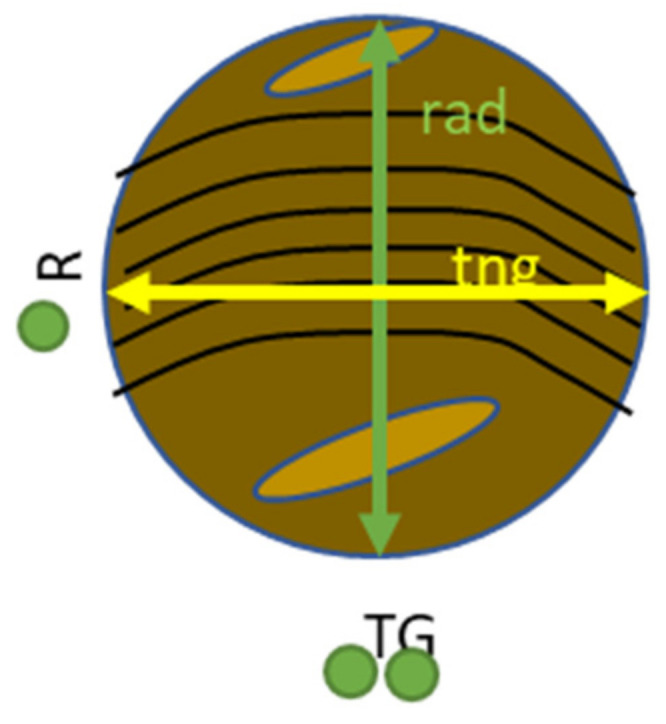
Directions of roughness measurement on the pin-on-disc test pin samples.

**Figure 14 polymers-14-01757-f014:**
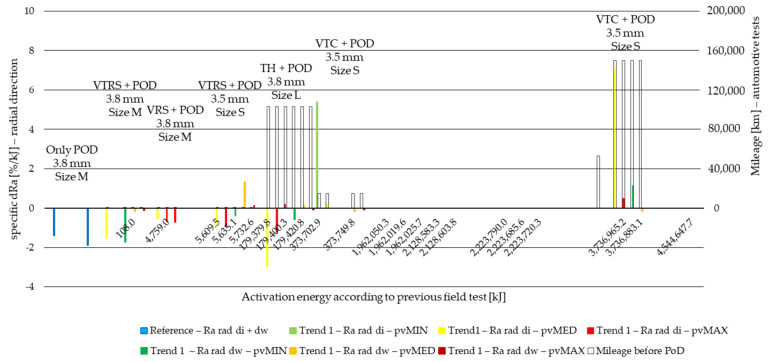
Radial direction PoD specific Ra difference values of cut material samples from inner and working diameter after minimum, medium and maximum *pv* level pin-on-disc tests with previous field-test (Table 2) mileage along Joule scale—Trend 1: clutch killer load cases: VTRS, VRS, TH, VTC.

**Figure 15 polymers-14-01757-f015:**
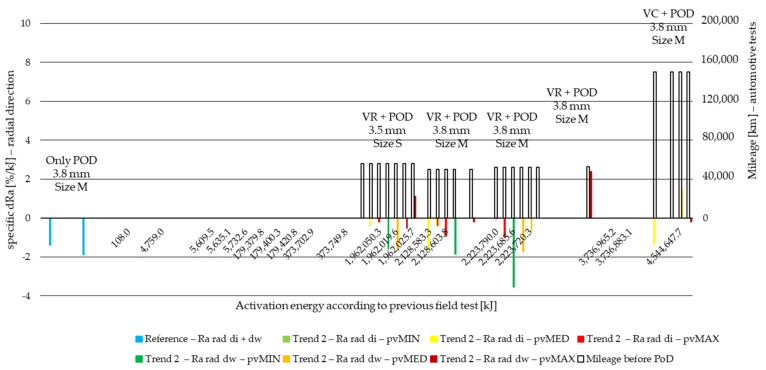
Radial direction PoD specific Ra difference values of cut material samples from inner and working diameter after minimum, medium and maximum *pv* level pin-on-disc tests with previous field-test (Table 2) mileage along Joule scale—Trend 2: moderate load cases: VR, VC.

**Figure 16 polymers-14-01757-f016:**
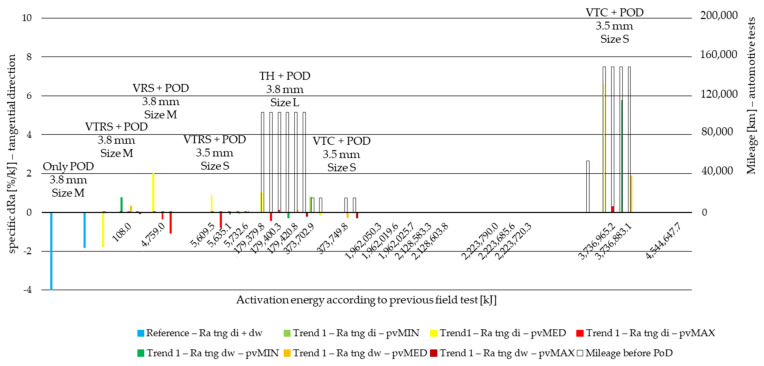
Tangential direction PoD specific Ra difference values of cut material samples from inner and working diameter after minimum, medium and maximum *pv* level pin-on-disc tests with previous field-test (Table 2) mileage along Joule scale—Trend 1: clutch killer load cases: VTRS, VRS, TH, VTC.

**Figure 17 polymers-14-01757-f017:**
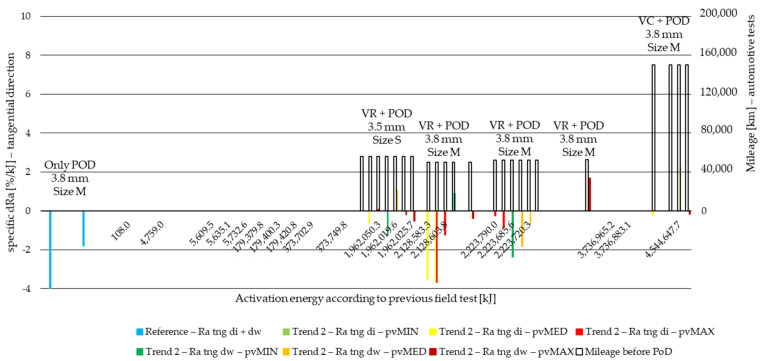
Tangential direction PoD specific Ra difference values of cut material samples from inner and working diameter after minimum, medium and maximum *pv* level pin-on-disc tests with previous field-test (Table 2) mileage along Joule scale—Trend 2: moderate load cases: VR, VC.

**Table 1 polymers-14-01757-t001:** Material properties of the identified and examined hybrid composite facing (adapted from [2]).

Property	Matrix	Fiber Reinforcement	Whole Material
Young modulus [MPa]	4290	27,300	direction dependent
Poisson’s ratio [–]	0.38	0.2	direction dependent
Shear modulus [MPa]	1290	11,380	direction dependent
Thermal conductivity coefficient [W/(m∙K)]	-	-	0.398

**Table 2 polymers-14-01757-t002:** Matrix of automotive tests: test samples along mileage and test intensity axes (adapted from [6]).

Samples from Automotive Tests				Mileage [1000 km]											
Test		Facing Size		0.006	0.5	0.8	15	39	45	50	52	53	56	102.9	150
Name	Test Intensity [J/km/cm^2^]	Diameters Outer/Inner [mm]	Thickness [mm]												
VRS	6.6	228/160	3.5												
		240/160	3.8			VRS-0008M									
		240/155													
VTRS	25	228/160	3.5		VTRS-0005S										
		240/160	3.8	VTRS-000M											
		240/155													
TH	55	228/160	3.5												
		240/160	3.8												
		240/155												TH-103L	
VC	120	228/160	3.5												
		240/160	3.8												VC-150M
		240/155													
VTC	128	228/160	3.5				VTC-015S								VTC-150S
		240/160	3.8												
		240/155													
VR	170	228/160	3.5										VR-056S		
		240/160	3.8					VR-039M	VR-045M	VR-050M	VR-052M	VR-053M			
		240/155													

## Data Availability

Not applicable.

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
