# Peer review of "Effects of Automotive Test Parameters on Dry Friction Fiber-Reinforced Clutch Facing Surface Microgeometry and Wear—Part 2"

_polymers, 2022, doi:10.3390/polym14091757_

Round 1
Reviewer 1 Report
Authors present a study about the effects of automotive test parameters on dry friction fiber reinforced clutch facing surface microgeometry and wear.
This is a second part of the article, where tribological stage is the main part of the publication.
This research manuscript may show interest to the audience of this journal. The abstract is concise but not sufficient; Tables and figures are appropriate but needs to be improved. Nevertheless, some observations, and changes are recommended. This manuscript can be considered for publication after addressing the following comments:
Comment 1. Some minor redaction mistakes have been detected, please take a moment to revise and correct them and English editing.
Comment 2. Abstract section needs to address some values regarding your main findings as well.
Comment 3. Conclusions section needs to address some values regarding your main findings as well.
Comment 4. Roughness is a factor of great influence on the frictional wear of surfaces, it is recommended to include surface roughness measurements prior to wear tests. Have you performed polishing processes on the pins after the cutting process?
Comment 5. The selection of the parameters described in table 2 needs to be justified.
Comment 6. The contact area of the pins and the disc may variate due to the cutting process. Have you ensured the perpendicularity of the surfaces of the tribological pair? Pleas describe that.
Comment 7. Wear rate of the tested specimens may be recommended to include.
Comment 8. Although ASTM G99 recommends evaluation by weight loss, it is also recommended by measuring wear groove dimensions. This is due to the fact that part of the worn material shows as adhesion and deformation. Therefore, it is recommended to complement the evaluation with wear groove measurements.
Comment 9. Error bars should be added to the results presented in the figures/graphs. How many repetitions of tests have you performs?
Comment 10. The conclusions only describe what is shown in the graphs, it is recommended to rewrite the conclusions providing conclusive hypotheses as to why the phenomena observed in the results occur.
Reviewer 2 Report
Here are my concerns.
> The literature review should complement the studies on testing methods and processes, in addition to the authors' previous studies.
>
> The impact of surface microgeometry should be more clearly highlighted and elaborated in the context.
>
> How do authors verify that the test condition conforms to the real working condition in vehicles?
Reviewer 3 Report
The article is well written and there are a couple of queries which must be addressed by the authors and they should make suitable modification to the manuscript accordingly
- Please include the tensile properties and fracture toughness of the woven fiber yarn composite material
- The authors can add the microstructures of worn surface and also analyze the wear debris to add few in-dept insights into the wear process of the fiber reinforced composite
Reviewer 4 Report
Dear Authors,
I have some comments on your article:
- At the end of the Introductions section, there is no information on how the article is organized.
- Literature should be checked if there are no newer items. Especially from the last 18 months. It would be good to add several references.
- It should be added in the summary whether the results of the work are implemented in the automotive industry.
Round 2
Reviewer 2 Report
No more comments.